# Occupational Difference in Oral Health Status and Behaviors in Japanese Workers: A Literature Review

**DOI:** 10.3390/ijerph19138081

**Published:** 2022-07-01

**Authors:** Koichiro Irie, Midori Tsuneishi, Mitsumasa Saijo, Chiaki Suzuki, Tatsuo Yamamoto

**Affiliations:** 1Department of Oral Health and Preventive Dentistry, School of Dentistry, Meikai University, Sakado 350-0283, Japan; saijo1218@gmail.com (M.S.); chiak@dent.meikai.ac.jp (C.S.); 2Japan Dental Association Research Institute, Chiyoda-ku 102-0073, Japan; tsuneishi_mi@jda.or.jp; 3Department of Dental Sociology, Kanagawa Dental University, Yokosuka 238-8580, Japan; yamamoto.tatsuo@kdu.ac.jp

**Keywords:** occupation, oral health status, oral health behavior, Japan

## Abstract

The occupational environment is an important factor for oral health because people spend a long time in the workplace throughout their lives and are affected by work-related stress and occupational health policies. This study aimed to review evidence for the association between occupation and oral health status and behaviors. A literature search of PubMed was conducted from February to May 2022, as well as a manual search analyzing the article origins. Articles were screened and considered eligible if they met the following criteria: (1) published in English; (2) epidemiological studies on humans; and (3) examined the association between occupation and oral health status and behaviors. All 23 articles identified met the eligibility criteria. After full-text assessments, ten articles from Japan were included in this review: four on the association between occupation and dental caries, three on occupation and periodontal disease, two on occupation and tooth loss, and one on occupation and oral health behaviors. An association was apparent between occupation, oral health status and behaviors among Japanese workers. In particular, skilled workers, salespersons, and drivers who work longer hours and often on nightshifts, tended to have poor oral health.

## 1. Introduction

The Japanese Ministry of Health, Labour and Welfare’s Survey of Dental Diseases and the first Health Japan 21 report indicate that oral health status in children has been improving in recent years, including a decrease in the prevalence of dental caries [1,2]. However, in the adult and older populations, the prevalence of dental caries and periodontal disease remains virtually unchanged or is increasing, despite increases in the numbers of teeth [1]. About 80% of all tooth loss is mainly the consequence of dental caries and periodontal disease. According to a survey conducted by the 8020 Foundation, tooth loss is rapidly increasing among adults [3]. Furthermore, dental caries and periodontal disease are among the most common chronic conditions worldwide. It is estimated that nearly 3.5 billion people have dental caries, periodontal disease, or another oral condition [4]. Therefore, oral conditions are a major burden in many countries and the means to prevent and improve oral health in adulthood is an urgent issue.

In addition, recent epidemiological studies have revealed that tooth loss exerts adverse effects on various systemic health conditions, including dementia and disability [5,6]. Because prevention methods for dental caries and periodontal disease have already been established, the most urgent issue in dental health care at present is how to efficiently deliver them to those in need.

Oral health is affected by many factors, such as socioeconomic status, cultural background, and nutritional habits [6,7,8]. Numerous studies suggested that occupational environment is a factor that affects oral health. Occupation has been used as a marker of socioeconomic status in several epidemiological studies [9,10,11]. Occupational environment is an important factor for health because people spend a long time at work during their lifetime. Therefore, work-related stress and health-care policies have a substantial effect on the oral health of older people [11].

To improve oral health status, especially among adults, the provision of dental health services in the workplace environment needs to be improved. Workplace oral health examinations accompanied by oral health instruction may also be effective for the maintenance of periodontal health. To help prevent and control oral disease, a public health approach that considers the workplace environment is urgently needed. Therefore, a literature search should be conducted to collect basic data regarding the relationship between occupation and oral health status and behaviors.

Therefore, to help prevent dental caries and periodontal disease, which both cause tooth loss, we aimed to collect scientific evidence and summarize the latest findings on the relationship between occupation and oral health status and behavior in the workplace using the following review question: Is occupation associated with oral health status and behaviors among workers? Furthermore, we also sought to identify which occupations are greater risk factors for poor oral health.

## 2. Materials and Methods

We conducted a literature review. Our research question for this review was: “Is occupation associated with oral health status and behaviors among workers?”

### 2.1. Eligibility Criteria

We set the following eligible criteria for literature screening. The following published studies were eligible: (1) those published in English; (2) epidemiological studies on humans; and (3) those that examined the association between occupation and oral health status and behaviors.

### 2.2. Information Source and Searches

Two reviewers (KI and MT) independently screened titles and abstracts for eligible studies. From February to May 2022, we searched for potentially relevant published studies using PubMed (published between 1996 and February 2022). We used the following search settings to obtain a wide range of studies: (“dental” AND “job”) OR (“oral” AND “job”) OR (“dental” AND “occupational”) OR (“oral” AND “occupational”) OR (“dental” AND “workplace”) OR (“oral” AND “workplace”). In addition, after one author assessed the titles and abstracts according to the aforementioned criteria, we also conducted a manual search of potentially suitable studies through the reference lists of the identified articles. All eligible studies were then selected for full-text reviews.

## 3. Results

Figure 1 presents the flow diagram of information through the phases of the review. The initial search identified 343 records. The titles of 320 were screened, and 23 articles met the eligibility criteria. After screening the abstracts, 13 were excluded for the following reasons: measured the effects on dental costs (*n* = 2); were about follow-up intervention oral health program (*n* = 5); were about oral symptoms (*n* = 2); were about oral and work performance (*n* = 3); were not about oral health status and occupation (*n* = 1). A total of 10 articles from Japan were finally included in this review.

### 3.1. Characteristics of Individual Studies

Table 1 shows summaries of all 10 articles. Four of the ten studies reported on the association between occupation and dental caries, three on occupation and periodontal disease, two on occupation and tooth loss, and one on occupation and oral health behaviors. The sample sizes among the included studies varied from 142 to 15,803.

### 3.2. Occupation and Dental Caries

In terms of the association between occupation and dental caries, three studies assessed dental caries using self-report online surveys and one as diagnosed by a dentist [12,13,14,15]. One cross-sectional study reported an odds ratio (OR) for dental caries that was 1.79 (95% confidence interval [CI], 1.20–2.67) times higher among males who worked nightshifts than those who worked dayshifts [15]. In addition, those who worked more overtime hours had a higher OR for dental caries as a result of neglecting dental care because of their busy work schedule [11,16]. Among females, the OR for dental caries was 3.51 (95% CI, 1.04–11.87) and 5.29 (95% CI, 1.39–20.11) times higher among those in professional/manager and service/sales occupations than among those who were homemakers or unemployed, respectively [15]. On the other hand, no significant difference was found between male gender and occupation.

### 3.3. Occupation and Periodontal Disease

Regarding occupation and the prevalence of periodontal disease, periodontal examinations were performed using the Community Periodontal Index. The results of a 5-year cohort study showed that among males, the relative risks for skilled workers, salespersons, and drivers were 2.52 (95% CI, 1.15–5.54), 2.39 (95% CI, 1.04–5.48), and 2.74 (95% CI, 1.10–6.79)-fold higher, respectively, than for other professionals [9]. By contrast, no significant difference was observed in females [9]. Furthermore, the results of a cross-sectional study showed that among males, the ORs for periodontal disease after adjusting for age, clinical history of diabetes mellitus and smoking status for drivers, service occupations, salespersons, and managers were 2.05 (95% CI, 1.66–2.54), 1.53 (95% CI, 1.19–1.96) and 1.49 (95% CI, 1.26–1.78), respectively, compared with other professionals [10]. Interestingly, the prevalence of periodontal disease also differed according to the number of employees, with those working in small- and medium-sized companies having a higher prevalence of periodontal disease than those working in large companies [11].

### 3.4. Occupation and Tooth Loss

In terms of occupation and tooth loss, two studies assessed tooth loss using self-report online surveys, and one study used a diagnosis by a dentist [10,17]. Drivers and employees working in manufacturing had fewer teeth than white-collar workers [11,17]. Those who worked at night also had a lower self-reported number of teeth than did those who worked during the day [12].

### 3.5. Occupation and Oral Health Behavior

In terms of occupation and oral health behaviors, differences in oral cleaning and dental visitation behaviors were found between occupations [18]. The percentage of those who did not brush their teeth before going to bed was higher among skilled workers. Furthermore, the percentages of salespersons who had a family dentist and received regular dental checkups were lower [18]. Taxi drivers had low numbers of teeth, which was significantly associated with diabetes, dietary and tooth brushing habits, and smoking status [17]. In addition, smokers accounted for 11.8% of those in the education and learning support category, 27.6% in the manufacturing industry, and 36.6% in the transport industry [11]. Considering individual factors such as education history and current economic condition, as well as regional factors such as urbanity, older adults who worked the longest as agricultural, forestry and fisheries workers had poor oral health status and behaviors, such as the infrequent use of an interdental brush or dental floss, compared with those in other occupations [18]. In addition, nightshift workers were less likely than dayshift workers to brush their teeth at least twice a day [14] and tended to have difficulty attending regular dental appointments [18]. 

## 4. Discussion

In this review, we summarized the existing literature on the association between occupation and oral health status and behaviors among workers. Based on the findings of this review, skilled workers, salespersons, and drivers, who work longer hours and often on nightshifts, tend to have poor oral health, such as a higher prevalence of dental caries and periodontal disease, and behaviors, thereby suggesting an association between occupation and oral health status and behaviors among workers. 

One possible mechanism for the differences in oral health status and behaviors among workers is that occupations that regularly involve long working hours and nightshifts tend to be associated with a lack of rest and adequate sleep, leading to higher work-related mental stress [19,20,21]. In particular, skilled workers, salespersons, and drivers tend to have longer working hours and often work nightshifts [9,16]. Taken together, this may be related social circumstances and psychosocial factors such as work-related mental demand and stress. Furthermore, a lack of flexibility in peoples’ daily lives reduces both the frequency and effectiveness of tooth cleaning [16,22]. These workers also tended to have difficulty attending regular dental appointments [18]. Therefore, the combination of occupational stress, time constraints, and fatigue caused by the work environment may influence poor oral health behaviors, resulting in differences in the prevalence of dental caries and periodontal disease among workers.

In addition, occupational classification is directly related to income and considers individual and contextual effects mainly based on the pathways linking income inequality and oral health [23]. To reduce oral health inequalities, the financial aspects of dental treatment should be considered [24,25,26]. For example, higher incomes are associated with important resources such as the ability to afford a higher quality diet and preventive and regular dental care. On the other hand, lower incomes are associated with poor oral health behaviors owing to higher rates of tobacco use and sugar consumption, infrequent and symptomatic dental visits, and poor oral hygiene [8,26]. Income inequality may be caused by lower social spending on public services and infrastructure, including dental care services [27]. Therefore, those in lower income brackets are more susceptible to tooth extraction, whereas those in with higher income brackets are more likely to seek periodontal care appointments and conservative dental treatment, resulting in a higher number of retained teeth [28].

Upstream contextual factors may also play an important role in the development of oral disease and oral health behaviors throughout the course of life, in connection with cultural and genetic factors from early stage of life. Recent research using the life-course framework suggested that exposure to adverse environments along the course of life additively contributes to oral health problems [29]. For example, the psychosocial environment in infancy and childhood influences oral health later in life. Adolescents at low-grade school levels are more likely to brush their teeth fewer than twice a day. Both slow educational progress and adverse socioeconomic and psychosocial environments are shown to be significantly associated with infrequent toothbrushing [30]. Therefore, the processes of accumulating advantages or disadvantages through a range of social experiences over time result in social inequalities in oral disease and behaviors that are observable in adulthood.

Some reports suggested that the occupational environment may be a risk factor for oral health status and behaviors, even when considering the various backgrounds of those engaged [9,10,11,15,16]. Therefore, adjusting these factors may be an effective method for preventing oral disease. To prevent and control dental caries and periodontal disease through the consideration of the occupational environment, a public health approach is needed. For example, workplace smoking-related measures are an important aspect of healthy workplace practices because smoking is a well-known risk factor for periodontal disease. Providing guidance that aims to encourage employees to stop smoking could be an effective strategy for preventing periodontal disease. Combined with relevant data, work environment inquiries may be useful for predicting and preventing dental caries and periodontal disease in those visiting the dentist for the first time or having a dental health checkup. In addition, this could be more important, not only for early detection through regular health checkups in the workplace, but also health education and guidance based on high-risk occupations in the future. 

Health education and guidance should be given priority in workplaces that mainly involve skilled workers, salespersons, and drivers. The present findings suggest that occupational status and shiftwork might be associated with the development of dental caries and periodontal disease. As these workplace issues might be potentially modifiable factors, the present review provides important information for dentists concerned with the prevention of dental caries and periodontal disease.

Our literature review has some limitations. In this review, all studies were conducted in Japan, although the literature search was conducted without restrictions by country. This might have resulted in an overestimation or underestimation due to sample bias, which may limit our ability to extrapolate the findings to the general population. Furthermore, well-designed cohort studies that include potential confounding factors, such as working hours, income level, job position, and oral hygiene practices, which are important determinants of oral health conditions, are needed.

## 5. Conclusions

The findings of the present review suggest the presence of an association between occupation and oral health status and behaviors among Japanese workers. In particular, skilled workers, salespersons, and drivers, who work longer hours and often on nightshifts, tended to have poor oral health status. 

## Figures and Tables

**Figure 1 ijerph-19-08081-f001:**
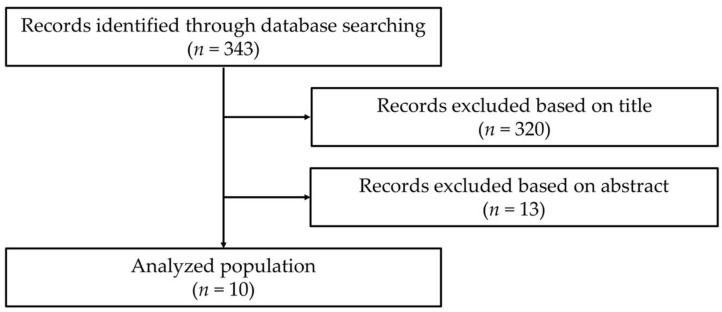
Flow of search strategy and selection of studies for a review.

**Table 1 ijerph-19-08081-t001:** Summary of included studies on occupation and oral health status and oral health behavior.

Reference No.	Author’s Name (Year of Publication)	Study Design	Exposure	Outcome	Number of Participants	Covariates	Main Results
**Dental caries**							
[12]	Ishizuka et al. (2016)	Cross- sectional	Daytime-only and night shift	Self-reported number of teeth present, reason for tooth loss, presence of untreated tooth	325 daytime workers and 351 night shift workers aged 30 to 69	Work schedule, age, household income, years of service, hypertension, smoking, habit, BMI, and daily brushing frequency	Tooth decay in night shift male worker (OR, 1.79; 95% CI, 1.20–2.67) from a multiple logistic regression analysis
[13]	Yoshino et al. (2017)	Cross- sectional	Over time work (0, over 0 to 45 h, over 45 to 80 h, over 80 h)	Self-reported untreated decayed teeth	951 male financial workers aged 25 to 64	Age, equivalent household income, educational background, overtime hours per month, brushing two times or more per day, eating between meals, having a regular dental clinic, interpersonal relationships in the workplace, and smoking habit	Tooth decay in 45–80 h of overtime work (OR, 2.56; 95% CI, 1.25–5.33) or over 80 h overtime work (OR, 3.01; 95% CI, 1.13–5.33) from a multiple logistic regression analysis
[14]	Ishizuka et al. (2019)	Cross- sectional	Self-reported questionnaire on the status of their own oral health	Self-reported untreated decayed teeth	142 male sales workers aged 30 to 49	Age, annual household income, eating between meals, night shift, and visiting a dental clinic in the past 6 months	Untreated tooth decay in working the night shift (OR, 3.492; 95% CI, 1.347–8.725) and visiting a dental clinic in the past 6 months (OR, 0.084; 95% CI, 0.010–0.733) from a multiple logistic regression analysis
[15]	Harada et al. (2018)	Cross- sectional	Occupational status (professionals and managers, clerical and related workers, service and salespersons, agricultural, forestry, and fishery workers, and homemakers and unemployed)	Presence of untreated decayed teeth	1342 workers (990 males and 352 females) aged 40 to 64	Gender, age, smoking status, and habit of eating sweets/drinking sweet drinks, BMI	Presence of untreated caries in female professionals and managers (OR, 3.51; 95% CI, 1.04–11.87) and service and salespersons (OR, 5.29; 95% CI, 1.39–20.11) from a multiple logistic regression analysis
**Periodontal disease**							
[9]	Irie et al. (2017)	Prospective Cohort	Occupational status (professional, managers, office workers, skilled workers, sales persons, service occupations, drivers)	Community Periodontal Index (a tooth scoring 3 or 4 indicates presence of periodontal disease)	3390 workers (2848 male and 542 female) aged 20 or over	Age, BMI, diabetes, smoking status, drinking status	CPI score of 4 in skilled workers (RR, 2.52; 95% CI, 1.15–5.54), sales persons (RR, 2.39; 95% CI, 1.04–5.48), and drivers (RR, 2.74; 95% CI, 1.10–6.79) compared with professional from a Poisson regression analysis
[16]	Morita et al. (2007)	Cross- sectional	Occupational status (professional, managers, office workers, skilled workers, sales persons, service occupations, drivers)	Community Periodontal Index (a tooth scoring 3 or 4 indicates presence of periodontal disease)	15803 male workers aged 20 to 69	Age, diabetes, smoking status	CPI score 4 in drivers (OR, 2.05; 95% CI, 1.66–2.54), service occupations (OR, 1.53; 95% CI, 1.19–1.96) and salespersons (OR, 1.49; 95% CI, 1.26–1.78) compared with professional from a multivariate logistic regression analysis
[11]	Zaitsu et al. (2017)	Cross- sectional	Industrial category, number of employees, job category, work schedule, and oral health behaviors	Decayed teeth, tooth loss and periodontal disease (CPI code 0–2 or 3–4),	1078 workers (808 male and 270 female) aged 19 to 70	Age, sex	Periodontal disease in company with fewer than 50 employees (OR, 15.56; 95% CI, 3.40–71.23) compared with 300 employees or more from a multiple logistic regression
**Tooth loss**							
[17]	Suzuki et al. (2016)	Cross- sectional	Occupational status (professional drivers and white collar workers)	Self-reported number of present teeth	592 professional drivers and 328 white collar male workers aged 30 to 69	Age, annual family income, working hours, shift work, duration of employment and night shift	Tooth loss in professional drivers (OR, 1.740; 95% CI, 1.150–2.625) compared with white collar workers from a multiple logistic regression analysis
[10]	Yamamoto et al. (2014)	Cross- sectional	Longest job (professional/technical, administrative, clerical, sales/service, skilled/labor, agriculture/forestry/fishery, others, no occupation)	Self-reported number of teeth, denture/bridge use, subjective oral health status and oral health behavior	23,191 subjects (11,310 males and 11,881 females) aged 65 or older	Age, educational attainment, equivalent income, densities of dentists and population	Having 19 or less teeth in agriculture/forestry/fishery workers (PR, 1.15; 95% CI, 1.06–1.26) compared with professional/technical from a multilevel Poisson regression analysis
**Oral health behavior**							
[18]	Suzuki et al. (2017)	Retrospective cohort	Self-reported associated with working environment	Regular dental appointment adherence rate	488 subjects (192 males and 296 females) aged 40 to 65	Age, sex and employment format	Regular dental attendance in night shift worker (OR, 0.220; 95% CI, 0.088–0.550) from a multiple logistic regression analysis

BMI, body mass index; CI, confidence interval; OR, odds ratio; PR, prevalence ratio; RR, relative risk.

## Data Availability

Not applicable.

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
