# Peer review of "Occupational Difference in Oral Health Status and Behaviors in Japanese Workers: A Literature Review"

_ijerph, 2022, doi:10.3390/ijerph19138081_

Round 1

Reviewer 1 Report

I suggest changing the title to "Association between Occupation and Oral Health Status and Behaviors among Japanese workers: A Literature Review"

I suggest mentioning Japan as a keyword and search item also this needs to be mentioned through out the MS

Remove the column with location as Japan from the table

Author Response

Comment 1: I suggest changing the title to "Association between Occupation and Oral Health Status and Behaviors among Japanese workers: A Literature Review"

Response to comment 1: Thank you for your professional comment. We agree with you and rewritten the title. 

Comment 2: I suggest mentioning Japan as a keyword and search item also this needs to be mentioned through out the MS. 

Response to comment 2: Thank you for your comment. We agree with you and rewritten the manuscript. 

Comment 3: Remove the column with location as Japan from the table.

Response to comment 3: Thank you for your comment. We agree with you and rewritten the table. 

Reviewer 2 Report

Dear Authors,

in attach I listed some remarks.

I hope you could find them useful to improve your paper.

Best regards

Author Response

Reviewer Comments to Author:

The topic of the article is interesting and the research is performed adequately. However I suggest to revise the article following the Prisma Statement. Some methodological aspects could be better described and reported in Materials and Methods. Results section could be improved with Figures reporting OR for each aspect considered. Below I listed some minor remarks that could improve the quality of your work.

Our comments: Thank you for your professional comment. We discussed about it with assistant Editor. As the type of this review, we considered the PRISMA protocol is not necessary.  We have added Figure about flow or search strategy.

  1. Line 18-20 please rewrite, it is not clear how many studies for each type of association you included.

Our comments: Thank you for your comment. We have added the sentence (Line 19-21).

  1. Line 83 correct ‘searched’ with ‘search’

Our comments: Thank you for your comment. We have revised the words. (Line 84).

  1. Line 52-62: the introduction to the impact of psychological stress in the workplace is weak and logical sequences in the text are not respected (‘rapid changes in global economy’ what stands for?) If you set up your phrase with this sentence, I prefer you consider in the introduction also the fact that in the period you performed the research of articles, the pandemic situation had also modified workplaces and therefore oral health behaviors during work.

Our comments: Thank you for your professional comments. We have added the sentences (Lines 55-60).

  1. Line 78, please specify the year you started the research

(May to February 2022)

Our comments: We have revised the words (Lines 79).

  1. Line 92 please rewrite. it is not clear how many studies for each type of association you included. You should list why you did not include 13 potential articles.

Our comments: Thank you for your comments. We have added the sentences (Lines 101-102).

  1. Discussion is the part of the article that needs a serious revision. You should focus mainly on the results of the articles you included in the reviews. Socioeconomic position of patients is not a variable studied in the articles included, therefore it should not be widely discussed. I suggest to add a paragraph regarding the limitations of this review and of the articles included. The eligibility criteria are generic and many aspects are not adequately considered as you correctly stated in Conclusions.

Our comments: Thank you for your professional comments. We have revised the sentences (Lines 185-187, 236-242, 247-248).

Reviewer 3 Report

Dear Authors, 

I have the following concerns regarding your manuscript. 

1. The title does not describe the nature of the review. Modify the title. 

2. Provide the Prospero registration number. 

3. The PRISMA flowchart and PRISMA check

4. Provide the ROB and summary bar plot figure. 

5. Provde the search strategy with results as a figure. 

Best Wishes. 

Author Response

1. The title does not describe the nature of the review. Modify the title. 

Our comment 1: Thank you for your comment. We have changed the title.

2. Provide the Prospero registration number. 

3. The PRISMA flowchart and PRISMA check

4. Provide the ROB and summary bar plot figure. 

Our comments 2-4: Thank you for your professional comment. We discussed about it with assistant Editor. As the type of this review (a literature review), we considered the PRISMA protocol is not necessary. Addition, we  considered Prospero and ROB is not necessary.  

5. Provde the search strategy with results as a figure. 

Our comments: Thank you for your comment. We have added Figure about flow of search strategy.

Round 2

Reviewer 1 Report

The title has not been corrected as mentioned before you need to add "Japanese workers"

This was a literature review and not a systematic review .. please remove the word systematic from the abstract

I think adding information about COVID 19 is irrelevant

The introduction needs to talk about the nature of Japanese workers like the amount of time spent and the challenges they face.

This will lead to my question... why studies only in Japan?

It needs to be more focused

Generalization of the results to other populations is questionable

This paper needs another round of revision

Author Response

Thank you for your professional comments.

According to the reviewer`s suggestions, we have revised the title and remove the word “systematic” in the abstract and sentenceregarding COVID 19 in the Introduction.

We did not add any information regarding Japanese workers, because this review did not limit to Japanesestudy. We did a literature search and happened to find only Japanese papers.

Regarding to comment about ` Generalization of the results to other population is questionable`, we have added the limitation in the Discussion section.

Reviewer 3 Report

Dear Authors, 

As far as its a literature review, I agree to skip ROB and PRISMA. Other comments are answered. 

Best Wishes

Author Response

Thank you for your suggestion. A native English speaker re-checked our manuscript.